# The Genetic Background Is Shaping Cecal Enlargement in the Absence of Intestinal Microbiota

**DOI:** 10.3390/nu15030636

**Published:** 2023-01-26

**Authors:** Silvia Bolsega, Anna Smoczek, Chen Meng, Karin Kleigrewe, Tim Scheele, Sebastian Meller, Silke Glage, Holger A. Volk, André Bleich, Marijana Basic

**Affiliations:** 1Institute for Laboratory Animal Science, Hannover Medical School, 30625 Hannover, Germany; 2Bavarian Center for Biomolecular Mass Spectrometry, TUM School of Life Sciences, Technical University Munich, 85354 Freising, Germany; 3Department of Small Animal Medicine and Surgery, University of Veterinary Medicine Hannover, 30559 Hannover, Germany; 4Center for Systems Neuroscience Hannover, 30559 Hannover, Germany

**Keywords:** germ-free animal models, genetic background, microbiota-dependent influences on host physiology, cecal enlargement, metabolism, digestive enzymes, mucins

## Abstract

Germ-free (GF) rodents have become a valuable tool for studying the role of intestinal microbes on the host physiology. The major characteristic of GF rodents is an enlarged cecum. The accumulation of mucopolysaccharides, digestion enzymes and water in the intestinal lumen drives this phenotype. Microbial colonization normalizes the cecum size in ex-GF animals. However, whether strain genetics influences the cecal enlargement is unknown. Here we investigated the impact of mouse genetic background on the cecal size in five GF strains frequently used in biomedical research. The cecal weight of GF mice on B6 background (B6J and B6N) represented up to 20% of total body weight. GF NMRI and BALBc mice showed an intermediate phenotype of 5–10%, and those on the C3H background of up to 5%. Reduced cecal size in GF C3H mice correlated with decreased water content, increased expression of water transporters, and reduced production of acidic mucins, but was independent of the level of digestive enzymes in the lumen. In contrast, GF B6J mice with greatly enlarged cecum showed increased water content and a distinct metabolic profile characterized by altered amino acid and bile acid metabolism, and increased acidic mucin production. Together, our results show that genetic background influences the cecal enlargement by regulating the water transport, production of acidic mucins, and metabolic profiles.

## 1. Introduction

Germ-free (GF) animal models represent a unique tool to study the function and impact of gut microbiota on the host under homeostatic or pathologic conditions. GF animals can be selectively colonized with specific microorganisms of interest or even microbial communities. Therefore, these models are frequently employed to decipher complex interactions between the host and its endogenous commensal microbiota, as well as to study interrelations between microbial species within the specific biotope such as the intestine [1,2]. The first GF animals were already generated by the late 19th century, and the first GF rodents were available in the 1940s [3,4,5,6]. Approximately 100 years later, with the increased interest in the microbiome research, GF animals were placed in the spotlight again. 

Life in the absence of intestinal microbiota is associated with various anatomical and physiological abnormalities [7]. The lack of microbial enzymes has a crucial impact on the metabolism of GF animals. In particular, the transformation of primary to secondary bile acids, as well as gut microbiota produced short-chain fatty acids and vitamins, are absent in GF animals [8,9,10,11]. Furthermore, GF animals have an underdeveloped immune system that is characterized by reduced gut-associated lymphoid tissue, low numbers of splenic and intestinal T helper cells, and the decreased production of intestinal immunoglobulin A due to reduced antigen priming [12,13,14]. However, the most prominent characteristics of GF rodents is an enlarged cecum [15,16]. Several factors that mediate this phenotypic characteristic of GF animals were proposed. The absence of intestinal microbes and their enzymes results in an altered osmolarity within the intestinal lumen. Mucopolysaccharides, which naturally undergo microbial degradation in the colonized gut, retain water and cause the dilatation of the cecum. Furthermore, the dilatation process is additionally supported by the low concentration of chlorine anions and intestinal atonia due to increased trypsin levels [15,17,18,19,20,21]. The colonization of GF rodents with intestinal bacteria normalizes the cecal size to the size observed in colonized animals within a few weeks [21,22]. Furthermore, the cecal size in GF rodents increases with age [16]. Greatly increased cecum can promote cecal torsion causing obstruction of the digestive tract followed by the partial constriction of the blood supply and the death of the animal with or without previous clinical symptoms such as abdominal distension, lack of feces, ragged fur, hypoactivity, and ataxia [23,24]. 

The contribution of the intestinal microbiota to the reduction of cecal dilatation is well documented. However, whether and how the host genetic background contributes to the degree of cecal enlargement is unknown. Therefore, in this study we assessed the cecal enlargement of all GF wild-type strains available at the Gnotobiotic unit of the Hannover Medical School. Five wild-type GF strains including the GF C57BL/6JZtm (B6J), C57BL/6NRjZtm (B6N), NMRI/MaxZtm (NMRI), Balb/cJZtm (BALBc), and C3H/HeOuJ (C3H) models frequently applied in biomedical research were analyzed to investigate the contribution of the host genetic factors on the cecal size in the absence of intestinal microbiota.

## 2. Material and Methods

### 2.1. Mice and Sample Collection

Adult germ-free (GF) male and female NMRI/MaxZtm (NMRI), C57BL/6JZtm (B6J), Balb/cJZtm (BALBc), C3H/HeOuJZtm (C3H), and C57BL/6NRjZtm (B6N) mice were obtained from the Central Animal Facility (Hannover Medical School, Hannover, Germany). Due to the breeding strategy of the gnotobiotic colonies, the NMRI mice are also classified as an inbred strain. GF animals were maintained and bred in plastic film isolators (Metall + Plastik GmbH, Radolfzell-Stahringen, Germany). The offspring of ex-GF B6J and C3H mice colonized with complex microbiota was used to assess the impact of microbial colonization on the host genetic factors. Complex microbiota colonized B6J and C3H mice were housed in individually ventilated cages under specific pathogen free (SPF) conditions. The environment of the animal rooms was controlled and standardized with 12-hour light/dark cycles. The health monitoring of SPF and GF animals was performed according to FELASA [25] and recommendations for maintaining gnotobiotic colonies [26], respectively. All animals used in this study were proven to be free of contaminants or infection with common murine pathogens except for *Helicobacter* sp., *Rodentibacter* sp., *Staphylococcus aureus*, *Pseudomonas aeruginosa*, and apathogenic intestinal flagellates in SPF C3H mice. Both GF and SPF rodents received pelleted 50 kGy gamma-irradiated feed (Complete feed for mice—breeding (M-Z), V1124-927), ssniff Spezialitäten GmbH, Soest, Germany) and autoclaved water ad libitum. At the age of 12 (±1) weeks, the mice were sacrificed by CO_2_ inhalation followed by cervical dislocation or exsanguination. Body weight, cecal weight and the weight of wet cecal content were recorded during the necropsy. To calculate the percentage of water and solid matter in cecal content, samples of wet cecal content were placed in the tube and incubated with open lids at 90 °C overnight. The next day the weight of the tube with dry cecal matter was recorded and the weight of the solid cecal matter was calculated. Additionally, cecal and colonic tissue (with fecal pellet) were collected for immunohistochemical and immunofluorescence staining. Blood was collected to perform a plasma analysis of liver and pancreatic enzymes. Cecal content, cecal and pancreatic tissue were collected, snap frozen in liquid nitrogen, and stored at −80 °C until further use. 

### 2.2. Ethical Statement

This study was conducted in accordance with German animal protection laws and with the European Directive 2010/63/EU. All procedures were approved by the local institutional Animal Care and Research Advisory committee and permitted by the Lower Saxony State Office for Consumer Protection and Food Safety or the local veterinary authorities (reference numbers: 42500/1H, 2018/188 and 19/3305). 

### 2.3. Quantitative Real-Time PCR (qPCR)

Total RNA was extracted from the cecum and pancreas for gene expression analyses using the RNeasy Kit (Qiagen, Hilden, Germany), including an additional step of on-column DNase digestion (RNase-Free DNase Set, Qiagen, Hilden, Germany). Afterwards, cDNA synthesis was carried out using the QuantiTect Reverse Transcription Kit (Qiagen, Hilden, Germany) according to the manufacturer’s recommendations. QuantiTect Primer Assays for claudin 2 (Mm_Cldn2_2_SG, Qiagen), claudin 4 (Mm_Cldn4_1_SG), claudin 8 (Mm_Cldn8_1_SG), mucin 2 (Mm_Muc2_2_SG), zonula occludens 1 (Mm_Tjp_1_SG), aquaporin 1 (Mm_Aqp1_1_SG), and aquaporin 4 (Mm_Aqp4_1_SG) were used to perform qPCR analyses. Beta actin (Mm_Actb_2_SG) served as an endogenous control. Gene expression analyses for facilitated glucose transporter (*Slc2a5*, Mm_00600311_m1), sodium/glucose cotransporter (*Slc5a1*, Mm_00451203_m1), and trypsinogen (*Prss2*, Mm_00657001_m1) were performed using TaqMan^®^ Gene Expression Assays and beta actin (Mm_00607939_s1) as a reference gene. Detection was performed with the QuantStudio 6 Flex Real-Time PCR System (Applied Biosystems, Weiterstadt, Germany) using the Fast SYBR Green^®^ Master Mix or TaqMan^®^ Fast Advanced Master Mix according to the manufacturer’s instructions. All reactions were run in triplicates. The amplified PCR product was verified by a melting curve analysis (for SYBR green chemistry). Samples were additionally normalized to the reference sample generated from mixed cDNA isolated from proximal colon, ileum, mesenteric lymph nodes, liver and spleen. Relative gene expression was calculated using the 2^−ΔΔCt^ method using the Thermo Fisher Connect^TM^ platform (Thermo Fisher Scientific, Waltham, MA USA).

### 2.4. Metabolomics

#### 2.4.1. Untargeted Metabolome Analysis

To assess the metabolic profile in the snap frozen cecal content of B6J and C3H mice, approximately 20 mg of cecal content was weight in a 2 mL bead beater tube (FastPrep-Tubes, Matrix D, MP Biomedicals Germany GmbH, Eschwege, Germany). The samples were homogenized and extracted with 1 mL of methanol using a bead beater (FastPep-24TM 5G, MP Biomedicals Germany GmbH, Eschwege, Germany) supplied with a CoolPrepTM (MP Biomedicals Germany, cooled with dry ice) for three cycles with 20 s of beating at a speed of 6 m/s and followed by a 30 s break. Afterwards, the suspension was centrifuged using an Eppendorf Centrifuge 5415R (Eppendorf, Hamburg, Germany) for 10 min at 8000 rpm at 10 °C. The clear supernatant was transferred to a 1.5 mL glass vial for measurement. 

An untargeted analysis was carried out on a Nexera UHPLC system (Shimadzu, Duisburg, Germany) connected to a Q-TOF mass spectrometer (TripleTOF 6600, AB Sciex, Darmstadt, Germany). Chromatographic separation was achieved by using an Acquity Premier BEH Amide 2.1 × 100 mm, 1.7 µm column (Waters, Eschborn, Germany) with a 0.4 mL/min flow rate. The mobile phase consisted of 5 mM ammonium acetate in water (eluent A) and 5 mM ammonium acetate in acetonitrile/water (95/5, *v*/*v*) (eluent B). The following gradient profile was used: 100% B from 0 to 1.5 min, 60% B at 8 min, 20% B at 10 min to 11.5 min, and 100% B at 12 to 15 min. Aliquots of 5µL per sample was injected into the UHPLC-TOF-MS. The autosampler was cooled to 10 °C and the column oven was heated to 40 °C. A quality control (QC) sample was pooled from all samples and injected after every tenth sample. The samples were measured in a randomized order and in the Information Dependent Acquisition mode (IDA). MS settings in the positive mode were as follows: Gas 1 55, Gas 2 65, Curtain gas 35, Temperature 500 °C, Ion Spray Voltage 5500, declustering potential 80. The mass range of the TOF-MS and MS/MS scans were 50–2000 *m*/*z* and the collision energy was ramped from 15–55 V. MS settings in the negative mode were as follows: Gas 1 55, Gas 2 65, Cur 35, Temperature 500 °C, Ion Spray Voltage -4500, declustering potential -80. The mass range of the TOF-MS and MS/MS scans were 50–2000 *m*/*z* and the collision energy was ramped from −15–−55 V. 

#### 2.4.2. Data Analysis

The “msconvert” from ProteoWizard [27] was used to convert raw files to mzXML (de-noised by centroid peaks). The bioconductor/R package xcms [28] was used for data processing and feature identification. More specifically, the matchedFilter algorithm was used to identify peaks (full width at half maximum set to 7.5 s). The peaks were then grouped into features using the “peak density” method [28]. The area under the peak was integrated to represent the abundance of features. The retention time was adjusted based on the peak groups presented in most samples. Missing values were first imputed using the “filling” methods provided by “xcms”, which integrated the intensity area according to the features’ retention time and mass range. The remaining missing values were then imputed with half of the limit of detection (LOD) methods; i.e., for very low abundant features, the missing values were replaced with half of the minimal measured value of that feature in all measurements. To annotate features with the names of metabolites, the exact mass and MS2 fragmentation pattern of the measured features were compared to the records in HMBD [29] and the public MS/MS spectra in MSDIAL [30], referred to as MS1 and MS2 annotation, respectively. The correctness of the MS2 annotation was manually reviewed and the features with valid annotations were considered in the downstream analysis. In the statistical analysis, the intensity between samples is median centered to normalize the effect of different loading volumes in the experiments. A *t*-test was used to compare the features’ intensity between B6J and C3H mice.

The Pathway Analysis module of MetaboAnalyst 5.0 [31] and Kyoto Encyclopedia of Genes and Genomes (KEGG) database for *Mus musculus* were used to analyze and visualize the affected metabolic pathways in B6J mice. For the pathway analysis, a list of annotated compounds upregulated in B6J was used. Compound name matching was performed to standardize the compound labels and not matched compound names were excluded from the subsequent pathway analysis. Results were calculated by the hypergeometric test and shown as the metabolome view.

The associated untargeted metabolomics data are available on the MassIVE repository (https://massive.ucsd.edu/ProteoSAFe/static/massive.jsp, accessed on 10 October 2022) with ID MSV000090765.

### 2.5. Immunohistology

Cecal tissue was collected and fixed in neutral buffered, 4% formalin for 24 h. Subsequently, samples were dehydrated, embedded in paraffin, sectioned at 3 μm, and stained with Alcian blue (AB, article no. 13416, Morphisto GmbH, Offenbach am Main, Germany) or periodic acid–Schiff (PAS, article no. 12153, Morphisto GmbH). The staining was performed as recommended by the manufacturer. For quantitative analysis, AB and PAS positive stained granules were counted per visual field (ten fields per section and animal). Sections were scored blindly using a Zeiss Axioskop 40 microscope (Carl Zeiss Microscopy GmbH, Göttingen, Germany) connected to an AxioCam MRm camera (Carl Zeiss Microscopy GmbH). The average size of the stained granules was measured using Zeiss ZEN blue software (4–5 animals per group, one representative image per animal including 25–74 measurements per image).

### 2.6. Immunofluorescence

Immunofluorescence staining for Ki-67 was performed on formalin-fixed paraffin embedded cecal tissue sections using rabbit anti-Ki-67 polyclonal primary antibody (1:200, Abcam, Cambridge, UK). Sections were first deparaffinized using xylol and rehydrated using decreasing concentrations of ethanol (100, 95, and 70%) followed by a short wash in distilled water. Heat-induced antigen retrieval was performed in a citrate-based buffer (Target Retrieval Solution, Agilent Dako, Santa Clara, CA, USA) in a 700 W microwave. Sections were blocked and permeabilized in PBS containing 10% horse serum and 0.1% Triton X-100 for 1 h at room temperature followed by overnight incubation at 4 °C with primary antibody. Primary antibodies were visualized by DyLight^®^594-conjugated donkey anti-rabbit polyclonal secondary antibody (1:250, Abcam) for 2 h at room temperature in the dark. PBS was used for washing. The number of Ki-67 positive cells was determined by counting the stained cells per visual field (ten fields per slide) and the mean value per animal was generated. Immunofluorescence staining for trypsin 2 was performed on Carnoy’s solution (60% absolute ethanol, 30% chloroform and 10% acetic acid) fixed and paraffin embedded colon tissue sections containing fecal pellet using rabbit anti-trypsin 2 polyclonal primary antibody (1:100; PA5-119849, Invitrogen, Waltham, MA, USA). Heat-induced antigen retrieval was performed in a Tris-EDTA Buffer (pH 9) in a pressure cooker for 40 min. Sections were blocked with 5% BSA in TBST for 30 min at room temperature. Staining was visualized with DyLight^®^488-conjugated donkey anti-rabbit polyclonal secondary antibody (1:500; Abcam) for 2 h at room temperature in the dark. Nuclear counterstaining was performed with a mounting medium containing DAPI (Vectashield, Vector Laboratories, Newark, CA, USA). Stained tissue sections were examined using the Zeiss Axioskop 40 microscope (Carl Zeiss Microscopy GmbH, Göttingen, Germany) connected to an AxioCam MRm camera (Carl Zeiss). All images were taken and scored blindly.

### 2.7. Qualitative Detection of Occult Blood in Intestinal Content

The qualitative detection of occult blood in cecal content was performed by using the Haemoccult^®^-test (Beckman Coulter GmbH, Krefeld, Germany). A lentil- to pea-sized sample of fresh cecal content was taken and applied on each of the two test fields. After drying for 5 days at room temperature, two drops of developer solution were centrally applied to each test field. Subsequently, the results (color changes) were analyzed within 30 to 60 s.

### 2.8. Determination of Digestive Enzymes in Cecal Content

#### 2.8.1. Quantitative Determination of Luminal Trypsin and Chymotrypsin in Cecal Content

Snap frozen cecal content samples were thawed on ice and homogenized in 0.9% sodium chloride (1 g sample/10 mL NaCl). The samples were then centrifuged at full speed for 20 min and the supernatant was taken for further analysis. Quantitative determination was performed as described previously, measuring the time needed for the drop of 0.1 pH unit in the solution [32]. Briefly, for trypsin analysis, calibration standards were created by diluting trypsin stock solution (1 mg/mL, T0303, Merck, Darmstadt, Germany) 1:2; 1:5; 1:10, 1:50, 1:100, 1:250, and 1:500 in trypsin buffer (0.354 g Tris-HCl, 0.334 g Tris, 2.34 g NaCl (0.04 M), and 2.94 g CaCl_2_ dihydrate (0.02 M) in 1L of distilled water, pH 8.2). A substrate solution was prepared by dissolving 2.075 g TAME (T4626, Merck) in a 50 mL trypsin buffer. For trypsin measurement, 2 mL of sample or standard solutions were mixed with 5.5 mL trypsin buffer and 2.5 mL substrate solution in a tube on a magnetic stirrer using a small metal staple as disposable stirrer. The pH of the solution was adjusted to just above 8.2 (with 0.2 M NaOH), and the time for pH decline from 8.2 to 8.1 was noted. For a chymotrypsin assay, enzyme stock solution (1mg/mL, C4129, Merck) was utilized for standard dilutions (1:2, 1:5, 1:10, 1:50, 1:100, 1:250, 1:500) in a chymotrypsin buffer (0.532 g Tris-HCl, 0.198 g Tris, 2.925 g NaCl (0.05 M), and 0.735 g CaCl_2_ dihydrate (0.05 M) in 1L of distilled water, pH 7.8). The substrate solution was prepared by dissolving 0.454 g ATEE (A6751, Merck) in 25 mL methanol and filled up to 50 mL with the chymotrypsin buffer. The analysis of luminal chymotrypsin was performed as described above, but a pH drop from 7.8 to 7.7 was recorded. Measurements were performed twice by readjusting the pH values of samples or standard mixtures and recording the time again using a pH-Meter 766 (Knick Labor, Germany). All measurements were conducted at room temperature. A standard curve was constructed by plotting time values and standard concentration values applying a log-log linear regression model. Enzyme concentrations were calculated from the standard curve and expressed as µg/mL. The chymotrypsin values for GF and SPF B6J mice were very low. The values that were outside the lowest range of the chymotrypsin standard curve were designated as 0 µg/mL chymotrypsin. 

#### 2.8.2. Trypsin Activity Assay

Trypsin activity in cecal content was detected using the colorimetric Trypsin Activity Assay Kit (ab102531, Abcam) according to the manufacturer’s recommendations, except for the optional step with the trypsin inhibitor sample. Supernatants of the samples used for the quantitative determination of luminal trypsin and chymotrypsin were further diluted 50-fold with 0.9% sodium chloride (the final dilution of the cecal samples for the assay was 1:500). Briefly, chymotrypsin inhibitor solution was added to each well containing a 50 µL sample and the positive control and was incubated for 10 min at room temperature. The reaction mix composed of the assay buffer and trypsin substrate was added to each well, and absorbance was immediately measured at 405 nm using a Multiskan SkyHigh Microplate Spectrophotometer (Thermo Scientific^TM^, Waltham, MA USA) in kinetic mode. 

### 2.9. Lifespan Analysis

Two aging cohorts of GF mice on a B6J and C3H background were followed over time to determine the average lifespan. Data in the curve show the end-points when GF mice were sacrificed. B6J mice were sacrificed when reaching humane end-points defined as the time-point when animal wellbeing was impaired. For the aging cohort, this was defined as the moment when the abdomen of the GF animals was visibly magnified (due to the greatly enlarged cecum) and started to interfere with the locomotor activity of the animal. The cecal enlargement was the major reason for sacrificing B6J mice, as the abdomen of GF mice dilated to the proportions that interfered with the locomotor ability of the affected mice. In contrast, mice on the C3H background were sacrificed at experimental endpoints. Not a single animal on the C3H background showed health issues related to cecal enlargement.

### 2.10. Liver and Pancreatic Enzyme Analysis from Blood

The blood samples were taken by puncturing the hearts of the euthanized mice. Whole-blood was transferred to lithium-heparin tubes and centrifuged at 7500× *g* for 5 min to extract plasma. Pancreatic and hepatic enzymes were measured in plasma using an analyzer for clinical chemistry (Cobas 4000 c311 analyzer, Roche Diagnostics GmbH, Mannheim, Germany).

### 2.11. Statistical Analysis

All data were analyzed using GraphPad Prism 6^®^ software (GraphPad Software, Version 6, La Jolla, CA, USA). The data were tested with a Shapiro-Wilk or Kolmogorov-Smirnov normality test for normal distribution. Parametric data were shown as mean + all points, and non-parametric data as median + all points. A t-test was used to statistically analyze the differences between two groups when data were parametric, and for non-parametric data the Mann-Whitney test was used. The statistical analysis of three or more groups was performed using a one-way ANOVA in the case of parametric data or a Kruskal-Wallis test with Dunn´s multiple comparison test in the case of non-parametric data. Pearson correlation calculations were performed using GraphPad Prism 6^®^ software with computing of the two-tailed *p* value. *p* < 0.05 was considered significant (* *p* < 0.05, ** *p* < 0.01, *** *p* < 0.001, **** *p* < 0.0001).

## 3. Results

### 3.1. Cecal Enlargement Varies between Different Germ-Free Strains

To determine whether the genetic background plays a role in cecal enlargement in the absence of intestinal microbiota, five wild-type germ-free (GF) strains including NMRI, BALBc, C3H, B6J and B6N were phenotypically analyzed. The cecum of GF mice takes up a large portion of the abdominal cavity (Figure 1A and Appendix A). The average body weight of 12 week old GF BALBc, C3H, B6J and B6N mice was around 25 g and comparable between strains. GF NMRI mice were heavier and showed an average body weight of 30 g. Furthermore, sex specific differences in body weight were observed in all analyzed GF strains, with males weighing more than females (Figure 1B). Cecal weight including cecal tissue and intestinal content was determined, and strain specific differences were observed (Figure 1C). GF mice on the B6 genetic background (B6J and B6N) had the largest cecal weight, and mice with C3H had the smallest cecal weight. Furthermore, the percentage of cecal weight of the body weight for each strain was determined (Figure 1D). In GF B6J and B6N mice, the cecal weight represented up to 20% of the total body weight. The cecal size in GF NMRI and BALBc mice was intermediate and represented between 5–10% of the total body weight. The cecum weight of GF C3H mice corresponded to approximately 5% of the total body weight. Interestingly, only GF B6J mice showed sex differences of the cecal enlargement, with males having a larger cecum than females (Figure 1D). As intestinal microbiota reduces the cecum size, we analyzed whether the differences in cecal size observed in the GF status were also present in complex microbiota colonized mice. To this end, the offspring of GF B6J and C3H mice born with complex intestinal microbiota was analyzed. As expected, the cecum of complex microbiota colonized mice was much smaller than in GF mice (Figure 1E and Appendix A). Complex microbiota colonized B6J and C3H mice similar to GF mice showed sex specific differences in body weight, with males weighing more than females. The average body weight of complex microbiota colonized C3H mice was higher than that of B6J mice (Figure 1F). However, when comparing the cecal weight and the percentage of cecal weight in total body weight between complex microbiota colonized B6J and C3H mice, mice with the C3H genetic background still had significantly reduced cecal size, but this difference was much less pronounced (Figure 1G,H). Moreover, not only the cecum size, but also the coloring of the cecal content was different between GF mice with larger and smaller cecum. Particularly in GF mice on the B6 genetic background (B6J and B6N), the color of the cecal content was dark brown to black (Figure 1A). As the dark color of intestinal content suggests the presence of blood, we analyzed the cecal content of GF B6J mice by the fecal occult blood test. However, no blood in cecal content of GF B6J mice was found (Figure 1I), indicating that the distinct coloring of the intestinal content is probably due to different metabolic properties. Altogether, these results show that the genetic background determines the phenotypic characteristics of the cecum. 

### 3.2. Host Genetic Background Shapes Distinct Metabolic Profiles in The Gf Mice Cecum

The intestinal content of GF mice contains increased water concentrations [15,16]. Therefore, we analyzed the influence of genetic background on the amount of water and solid matter in the cecal content. To analyze this, wet cecal content was collected and dried to determine the percentage of water and solid matter. The amount of water correlated with the cecal weight (Appendix A). The water content was the highest in GF B6J and B6N mice, intermediate in GF NMRI and BALBc mice, and the lowest in GF C3H mice (Figure 2A,B). The amount of solid matter was inversely correlated to the cecum size, as GF mice with B6 genetic background had the lowest levels and GF C3H the highest levels of cecal solid matter (Figure 2B,C). Cecum size increases with the age of the animal and it can lead to the twisting of the cecum, when greatly enlarged, and cause intestinal obstruction followed by the death of the animal. To assess the impact of the cecal size on the average lifespan of GF mice, we followed two aging cohorts of mice on the B6J and C3H genetic background. The average lifespan of GF B6J mice was shorter than that of GF mice with the C3H background (Figure 2D). In contrast to GF mice with the C3H genetic background, GF B6J mice reached humane end-points at around one year of age. At that age, GF B6J mice were euthanized primarily due to health issues induced by the enlarged cecum. Due to the significantly enlarged cecum, the abdomen of B6J mice was significantly larger, and this influenced the locomotor activity of these mice. The C3H aging cohort did not reach a humane end-point due to health issues associated with the enlarged cecum, and was sacrificed at experimental end-points. Furthermore, we also analyzed the composition of cecal content of complex microbiota colonized B6J and C3H mice. The presence of intestinal microbiota strongly reduced the percentage of water and increased the amount of solid matter in cecal content (Figure 2E–G). No difference in water and solid mater in the intestinal content was observed between colonized B6J and C3H mice, supporting the strong impact of intestinal microbiota on the host phenotype.

Next, we analyzed the metabolic profiles of GF B6J and C3H mice using a mass spectrometry-based untargeted metabolomics approach. Combining positive and negative ionization mode, 2402 metabolic features were detected in total (Appendix A). The principal component (PC) analysis clearly separates the B6J and C3H mice on PC1, accounting for over 40% of total variance in the data (Figure 2H,I). This result indicates the distinct metabolic profile of host- and food-derived metabolites in the two strains. From a total of 2402 metabolic features, 310 (106 in negative and 204 in positive ion mode) were significantly altered between GF B6J and C3H (*t*-test, false discovery rate). Quite a proportion of detected metabolic features were poorly or not annotated, so we focused on the analysis of well-annotated features based on their respective MS1 and MS2. This finally included 22 (17 in positive and 5 in negative ion mode; Figure 2J, Appendix A) distinctly regulated metabolic features between GF B6J and C3H mice. In addition, the metabolites genistein, tauro-α-muricholic acid, L-proline, L-threonine, L-glutamine, L-glutamic acid, thymidine, and diferuloyl putrescine were detected in both ion modes. A hierarchical clustering analysis was performed on these metabolic features, and they were visualized using a heat map (Figure 2J). In addition, their expression profiles were shown using beeswarm plots (Appendix A). The changes in metabolic profiles between GF B6J and C3H mice of these well-annotated metabolites involved changes in the metabolism of amino acids, primary bile acids and plant-derived compounds (Figure 2J). In contrast to GF C3H mice, GF B6J mice showed a higher abundance of amino acids such as L-proline, L-glutamine, L-glutamic acid, L-arginine and L-threonine in their cecal content. Changes in amino acid metabolism were also supported by metabolic pathway analysis using a pathway-based analysis tool (Figure 2K). One of upregulated metabolites found in GF B6J mice was also L- pyroglumatic acid. This is particularly interesting as the sodium salt of pyroglutamic acid displays water-absorbing properties, indicating that the increased water in the cecal lumen of B6J mice could be due to the presence of water-absorbing substances. Besides changes in the amino acid metabolism, GF B6J mice also had a higher abundance of tauro-α-muricholic acid, which is a taurine-conjugated form of the murine-specific primary bile acid α-muricholic acid. The distinct expression profiles of diferuloyl putrescine (phenolamide) and sissotrin (isoflavone) in GF C3H, as well as genistein (isoflavone), O-malonylgenistin (isoflavone) and α-tocopherol acetate (synthetic form of vitamin E) in GF B6J mice, indicate that mice with different genetic backgrounds differently metabolize food-derived compounds. Moreover, GF B6J mice also showed a higher expression of a lysophospholipid LysoPC 16:0 in the cecal content, suggesting that lipid metabolism pathways also differ between these two strains.

### 3.3. Reduced Cecal Size Correlates with Higher Water Transportation across the Epithelial Barrier

As cecal enlargement was associated with increased water content in GF mice, and we next assessed the two main water transport pathways. Water can be transported across the gut epithelium by the paracellular (between cells) and transcellular (through the cell) routes [33]. The paracellular route involves a passive transport process regulated by a tight junction. Thus, we investigated the expression of tight junction genes Zonula occludens 1 (ZO1), claudin 2 (pore-forming claudin), and claudin 4 and 8 (sealing claudins). The gene expression of tight junctions was comparable between all five GF strains with minor genetic background-induced differences in gene expression of ZO1 and claudin 4 (Figure 3A–D). Furthermore, we also assessed additional barrier determining factors such as mucin 2 expression and epithelial cell proliferation. GF B6J mice showed higher mucin 2 gene expression than GF B6N mice. Mucin 2 gene expression was also higher in GF BALBc when compared to GF NMRI and B6N mice (Figure 3E). Epithelial cell proliferation was determined by the staining of cecal tissues with Ki67 proliferation marker. GF mice on the B6 genetic background showed slightly reduced epithelial cell proliferation. However, no statistically significant difference between all five GF strains was detected (Figure 3F,G). These results indicate that increased paracellular permeability regulated by the gene expression of tight junctions is not responsible for observed differences in water content between the five GF mouse strains. However, different distributions of these proteins alongside the intestinal barrier that is independent of changes in gene expression cannot be excluded with this analysis as a potential factor in regulating paracellular water transport. As water is also cotransported with sodium and sugar, we next analyzed the gene expression of two solute transporters: the facilitated glucose transporter (*Slc2a5*/*Glut5*) and the sodium/glucose cotransporter (*Slc5a1*/*Sglt1*). The gene expression of both solute transporters in cecal tissue was significantly upregulated in GF C3H mice compared with GF mice on the B6 genetic background (Figure 4A,B). Additionally, the gene expression of facilitated glucose transporter *Slc2a5* was also higher in GF C3H mice in comparison to GF BALBc and NMRI mice, indicating higher water absorption from cecal lumen in GF C3H mice. In addition, a Pearson correlation analysis showed that there is a negative correlation between the water content and the *Slc5a1* and *Slc2a5* gene expression (Appendix A). 

Water can also be transported through aquaporins (AQPs), a family of water-channel proteins that are expressed on intestinal epithelial cells [34]. Thus, we also determined the gene expression of AQP1 and AQP4 in the cecal tissue. The gene expression of AQP1 was comparable between all strains (Figure 4C). The gene expression of AQP1 was upregulated only in GF C3H mice in comparison with GF NMRI mice. Additionally, GF B6N mice also showed increased AQP1 gene expression when compared with GF BALBc mice (Figure 4C). The gene expression of AQP4 in cecal tissue showed more pronounced genetic background differences. GF C3H mice showed significantly upregulated AQP4 gene expression in comparison with GF B6J, B6N and NMRI mice. In addition, GF BALBc mice also had a higher gene expression of AQP4 in comparison with GF NMRI and B6J mice. However, no correlation between AQP4 gene expression and water content was determined (Appendix A). Altogether, these data indicate that the gene expression of solute and water transporters is increased in GF C3H mice, suggesting a higher water absorption from the cecal lumen in these mice. 

Furthermore, we also analyzed the microbial influence on solute transporters and AQPs gene expression in complex microbiota colonized B6J and C3H mice. The difference in the gene expression of glucose transporter *Slc2a5* observed in GF B6J and C3H mice were eliminated in colonized mice, indicating that the gene expression of these genes can be regulated by intestinal microbiota (Figure 4E). Interestingly, the difference in the gene expression pattern of the sodium/glucose cotransporter between GF C3H and B6J mice was also observed in complex microbiota colonized mice, indicating that the gene expression of *Slc5a1* is at least partially genetically determined (Figure 4F). The colonization status did not change the gene expression pattern of AQP1 between microbiota colonized C3H and B6J mice (Figure 4G). The gene expression of AQP4 did not differ between C3H and B6J microbiota colonized mice (Figure 4H), indicating that the expression of this gene can potentially be influenced by intestinal microbiota. Altogether, these data suggest that the decreased expression of solute/water transporters and water channels could be responsible for water accumulation in B6J mice.

### 3.4. GF B6J Display Increased Production of Acidic Mucins in the Gut

Accumulated mucins in the cecal lumen that attract water are described as one of the factors contributing to the development of the cecal enlargement in GF rodents [18]. To analyze whether the differences in cecal size between GF B6J and C3H mice is due to the distinct production and secretion of intestinal mucins, cecal tissues were stained with Alcian blue (AB) and periodic acid—Schiff (PAS) techniques. Alcian blue staining at a pH of 2.5 stained acid mucins deep blue (Figure 5A,B). The quantification of AB positive cells in the cecal tissue of GF B6J and C3H mice showed that the number of AB positive cells in GF B6J was significantly higher than in GF C3H mice (Figure 5C). Moreover, we also compared the average size of AB stained granules. However, there was no difference in the size of acidic mucin granules between GF B6J and C3H mice (Figure 5D). When comparing the number of AB positive cells as well as the average size of AB stained granules between complex microbiota colonized B6J and C3H mice, no difference was observed. However, in comparison with GF mice, only complex microbiota colonized C3H mice showed significantly higher numbers of AB positive cells as well as average granule size than their GF counterparts (Figure 5C,D). Similar analyses were performed with PAS staining. PAS staining stained neutral mucins from pink to red (Figure 5E,F). The quantification of PAS stained cells in the cecal tissue of GF B6J and C3H mice revealed no difference in the number of PAS positive cells (Figure 5G). In addition, no difference in the average size of PAS stained granules between GF B6J and C3H mice was observed (Figure 5H). Furthermore, similar to AB staining, no difference between PAS positive cells and their granule size was observed between complex microbiota colonized B6J and C3H mice (5G–H). However, the average size of PAS granules was only significantly increased in complex microbiota colonized C3H mice when compared to GF C3H mice (Figure 5H). As AB and PAS stained granules were bigger in complex microbiota colonized mice, this finding indicates that the intestinal microbiota composition could be responsible for this phenotypic observation. Altogether, these data show that GF B6J mice produce and secrete more acidic mucins in the gut lumen that in turn can retain more water and influence the cecal enlargement. 

### 3.5. Reduced Cecal Size in Gf C3h Mice Is Independent of Increased Digestive Enzyme Levels

The large intestine of GF mice contains increased levels of digestive enzymes, the accumulation of which could also be responsible for the cecal dilatation [35]. Thus, we hypothesized that GF C3H mice in comparison with GF B6J mice show decreased levels of digestive enzymes. As both the liver and pancreas support digestion processes, we first analyzed liver and pancreatic enzymes in the plasma of both GF and complex microbiota colonized B6J and C3H mice. The parameters for assessment of liver function involved the analysis of alkaline phosphatase (ALP), alanine aminotransferase (ALT), aspartate aminotransferase (AST), and bilirubin. Plasma levels of ALP (Figure 6A), AST (Figure 6B) and bilirubin (Figure 6C) were comparable between GF and complex microbiota colonized B6J and C3H mice. Only GF C3H mice showed decreased plasma levels of ALT in comparison with GF B6J mice (Figure 6D). The pancreas produces three types of digestion enzymes: lipase, amylase and protease. Pancreatic lipase and amylase are secreted as active enzymes. The level of pancreatic lipase and amylase were thus determined in the plasma. GF and complex microbiota colonized mice on the C3H background secreted decreased levels of pancreatic lipase in comparison with GF and complex microbiota colonized B6J mice, indicating a strain-specific characteristic (Figure 6E). No differences in the plasma level of amylase were observed between GF and colonized B6J and C3H mice (Figure 6F). Furthermore, all pancreatic digestive proteases are secreted as inactivated enzymes. Therefore, to analyze whether GF B6J mice secrete more trypsinogen than GF C3H mice, we examined its production in the pancreas by measuring the gene expression of serine protease 2 (*Prss2*). No difference in the gene expression of trypsinogen in the pancreas was observed between GF and colonized B6J and C3H mice (Figure 6G). We then analyzed the levels of activated digestive proteases trypsin and chymotrypsin in the cecal content. Both GF B6J and C3H mice displayed high levels of luminal trypsin. However, no differences between luminal trypsin activity (Figure 6H), concentration (Figure 6I) and immunostaining (Figure 6J) were detected between GF B6J and C3H mice. In comparison with GF B6J and C3H mice, complex microbiota colonized B6J and C3H mice had substantially lower levels of luminal trypsin (Figure 6H–J). Luminal chymotrypsin levels were elevated in GF C3H mice in comparison with GF B6J (Figure 6K). Furthermore, luminal chymotrypsin levels were reduced in complex microbiota colonized C3H mice and were mostly below the detection level in both GF and microbiota colonized B6J mice (Figure 6K). Altogether, these data suggest that the reduced cecal size in GF C3H mice is not mediated through decreased levels of digestive enzymes in the intestinal content.

## 4. Discussion

The intestinal microbiota has an immense influence on host physiology and metabolism. The absence of intestinal microbial communities causes anatomical and physiological changes in living organisms. The most representative characteristic of germ-free (GF) rodents is the highly dilated cecum in response to the sterile environment. This specific phenotype was described as early as the first GF guinea pigs were generated [3]. The role of intestinal microbiota on the cecal size is well described. The colonization of GF animals reduces the cecal size to the one observed in colonized animals, whereas the antibiotic treatment of complex microbiota colonized animals increases the cecum size [11,22,36,37,38]. However, whether the host genetics influences the cecal enlargement is unknown. Here, we show that the strain background genetics determines the proportion of the cecal dilatation. In this study, we studied the cecal enlargement in five GF wild-type strains that are used in biomedical research. We showed that the cecal weight in GF B6J and C3H mice represented 20% and 5% of the total body weight, respectively. In addition, in two other strains, BALBc and NMRI, the cecal weight represented between 5 and 10% of the total body weight. These results confirmed that the host-mediated factors also shape the cecal enlargement in the absence of intestinal microbiota. The factors that were implicated in the cecal enlargement besides the presence of microorganisms are the increased secretion of water and electrolytes, the accumulation of mucins, undigested fibers, and host-mediated bioactive substances, together with decreased gut motility [15,17,18,39,40,41,42]. As GF C3H mice had reduced cecal size in comparison with GF B6J mice, we hypothesized that this is due to reduced water transportation into the lumen and the lower amount of undegraded host-derived substances such as mucins and dietary enzymes. The water content in the cecal lumen of GF C3H mice was lower than in GF B6J mice, and this was associated with the increased gene expression of solute transporters *Slc2a5* and *Slc5a1*, as well as aquaporin 4. As water is cotransported with sodium and sugars across the intestinal epithelium, this suggests increased water absorption in GF C3H mice [43,44]. Additionally, previous studies showed that dietary nonabsorbable polymer polyethylene glycol induces the hypertrophy of the rat cecum and stimulates sodium transport [45,46]. In addition, aquaporins (AQPs), water-channel proteins that are expressed throughout the gastrointestinal tract, are involved in water transport across the epithelium and play a role in the dehydration process of fecal contents [34,47]. The decreased expression of AQP4 was associated with the induction of allergic or infection-related diarrhea and intestinal inflammation [48,49,50]. Previous findings suggested that the water retention in the cecal lumen of GF animals is due to the presence of osmotically active substances [51]. This is in line with our metabolomic data that show the increased presence of L- pyroglumatic acid (PCA) in the cecal lumen of GF B6J mice. The PCA, a naturally occurring component of mammalian tissue, and its derivates are classified as humectants (water-absorbing substances) [52]. GF B6J mice also showed significantly higher levels of taurine-conjugated primary bile acids in the cecal lumen. As bile acids can also inhibit water transport and motility in the large intestine, their increased levels in B6J could contribute to the greatly enlarged cecum [19,53,54]. The difference in the primary bile acid concentration points to an endogenous effect and is not dependent on the food. Besides these differences, GF B6J and GF C3H mice differently metabolize food-derived components independently of the intestinal microbiota. The metabolism of plant-derived genistein, an isoflavone, differs between these two strains. The naturally occurring form as well its glycoside form, O-malonylgenistin, were more expressed in the cecal content of GF B6J mice, whereas in GF C3H mice a methylated and glycosylated form of genistein, sissotrin, was found. Distinct metabolic profiles also involved plant-derived phenolamide, diferuloyl putrescine, and the diet supplemented synthetic form of vitamin E, α-tocoperol acetate, expressed more in GF C3H and GF B6J mice, respectively. This supports the findings that the genetic background influences phenotypic properties [55,56,57]. Additionally, these two GF strains also showed changes in the amino acid metabolism. GF B6J mice had increased expression profiles of certain amino acids such as L-proline, L-glutamine, L-glutamic acid, L-arginine and L-threonine in their cecal lumen. These amino acids were attributed to different pathways with the highest impact on the glutamine and glutamate metabolism. Glutamine and glutamate are important for the maintenance and promotion of the cell functions such as serving as an energy source, promoting enterocyte proliferation, tight junction regulation, and the suppression of pro-inflammatory signaling [58,59]. As the difference between digestive enzymes between GF B6J and GF C3H mice were not significant, the higher level of amino acids in the lumen of GF B6J could potentially be due to their reduced absorption. Water uptake influences the absorption of amino acids from the lumen [60,61]. As GF B6J mice have reduced water transport and more water in the lumen than GF C3H mice, this could influence the difference in the amino acid levels between these two strains. The accumulation of unabsorbed and partially undigested proteins in the cecum such as mucins and dietary enzymes has also been proposed as a contributing factor to the cecum dilatation. Mucins are large, highly glycosylated proteins that are categorized into neutral and acidic subtypes that build up the mucus layer in the intestine [62,63]. The mucin glycans are hydrophilic and bind water to form a gel-like mucus layer that protects epithelial cells from dehydration and mechanical stress [64]. The accumulation of mucins in the GF lumen was suggested to be due to the absence of mucus-degrading bacteria rather than the increased production of mucins [62]. The gene expression of the mucin 2 gene, the main component of the mucus in the large intestine, was not altered between GF B6J and C3H mice. However, we showed that GF B6J mice have an increased number of acidic mucin producing cells in the intestinal mucosa, suggesting that the higher production of acidic mucins could contribute to the increased cecal size in these mice. Dietary enzymes are also not absorbed by the intestinal epithelium and thus accumulate in the lumen of GF animals [19]. We showed that GF C3H mice produce higher levels of luminal chymotrypsin and reduced levels of lipase in the blood. The increased lipase activity in GF B6J mice could be due to the increased expression of LysoPC 16:0 in the cecal content, as detected by metabolomic analyses. No difference in the trypsin activity has been identified between GF B6J and C3H mice, supporting the hypothesis that the accumulation and production of dietary enzymes is not mediating the difference in cecal size between these two strains. We also confirmed marked differences in the trypsin levels between GF and complex microbiota colonized mice. This is in line with the recent publication in which specific intestinal taxa responsible for trypsin degradation in the large intestine was identified [65]. Immunohistological trypsin staining supported the proposed mechanism of trypsin degradation by trypsin recruitment to the surface of commensal bacteria, as bacteria-like structures in the lumen were stained with trypsin antibody. GF animals were shown to live longer than mice on the same genetic background colonized with intestinal microbiota, and do not develop signs of inflammaging [66,67]. However, a greatly enlarged cecum can cause health issues in aged GF animals. Cecal torsion due to a greatly enlarged cecum was described as the major health issue in aged GF animals [20,24]. Using two aging cohorts, we showed that the expected lifespan of GF B6J is shorter than that of GF C3H mice. GF B6J mice were sacrificed at approximately 1 year of age, as their abdomens became significantly larger due to the enlarged cecum, and this led to locomotor difficulties. In addition, we did not observe any cecal twisting in these cohorts. However, due to large differences in the lifespan, the application of GF mice on a C3H background could be recommended for studies involving aged GF mice.

Altogether, we showed that the host genetic background determines the extent of the cecal enlargement in the absence of intestinal microbiota by regulating the water transport across the epithelial barrier, the production of acidic mucins in the cecal mucosa, and by shaping distinct metabolic profiles.

## Figures and Tables

**Figure 1 nutrients-15-00636-f001:**
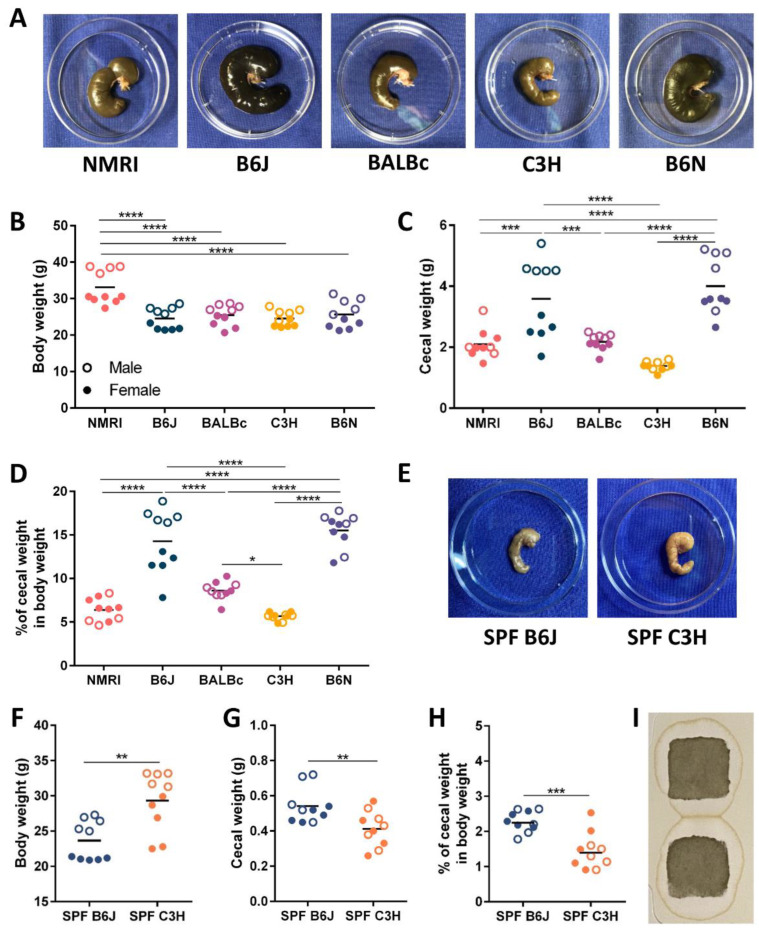
The genetic background influences the cecal size in mice. **(A**) Representative photographs of the cecal size in five GF strains (NMRI, B6J, BALBc, C3H and B6N; n = 10). (**B**) Total body weight of five GF strains (NMRI, B6J, BALBc, C3H and B6N) (open circles (○) males, closed circles (●) females). Statistical differences were calculated by one-way ANOVA with Tukey’s post-hoc test **** *p* < 0.0001. Each dot represents one independent sample. (**C**) Cecal weight (including tissue and luminal content) of five GF strains (NMRI, B6J, BALBc, C3H and B6N) (open circles (○) males, closed circles (●) females). Statistical differences were calculated using one-way ANOVA with Tukey’s post-hoc test *** *p* < 0.001, **** *p* < 0.0001. Each dot represents one independent sample. (**D**) Percentage of cecal weight in body weight of five GF strains (NMRI, B6J, BALBc, C3H and B6N). Statistical differences were calculated using one-way ANOVA with Tukey’s post-hoc test **** *p* < 0.0001, * *p* < 0.1. Each dot represents one independent sample. (**E**) Representative photographs of the cecal size in complex microbiota colonized B6J and C3H mice. (**F**) Total body weight of complex microbiota colonized B6J and C3H mice (open circles (○) males, closed circles (●) females). Statistical differences were calculated by unpaired *t*-test ** *p* < 0.01. Each dot represents one independent sample. (**G**) Cecal weight (including tissue and luminal content) of complex microbiota colonized B6J and C3H mice (open circles (○) males, closed circles (●) females). Statistical differences were calculated by unpaired *t*-test ** *p* < 0.01. Each dot represents one independent sample. (**H**) Percentage of cecal weight in body weight of complex microbiota colonized B6J and C3H mice (open circles (○) males, closed circles (●) females). Statistical differences were calculated by unpaired *t*-test *** *p* < 0.001. Each dot represents one independent sample. (**I**) Representative picture of negative Hemoccult test of cecal content from GF B6J mice (n = 4). Abbreviations: GF, germ-free; SPF, specific pathogen free.

**Figure 2 nutrients-15-00636-f002:**
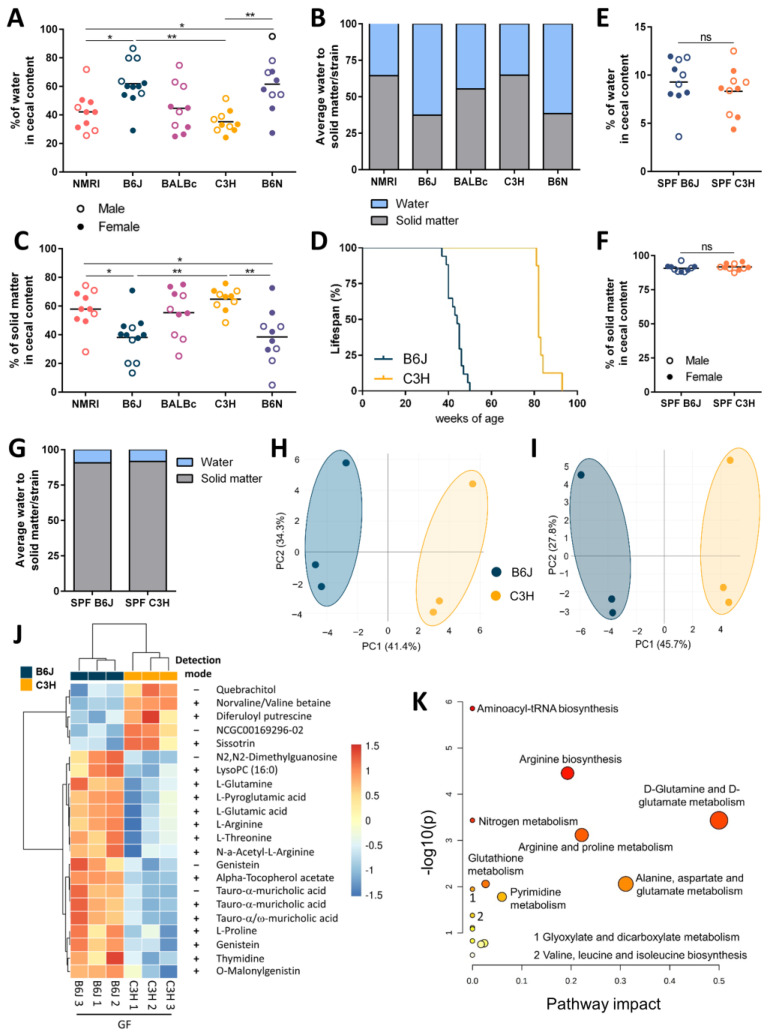
The genetic background is shaping distinct metabolic profiles. (**A**) Percentage of water in the cecal content of five GF strains (NMRI, B6J, BALBc, C3H and B6N) (open circles (○) males, closed circles (●) females). Statistical differences were calculated by one-way ANOVA with Tukey’s post-hoc test * *p* < 0.05, ** *p* < 0.01. Each dot represents one independent sample. (**B**) Average water to solid matter in the cecal content of five GF strains (NMRI, B6J, BALBc, C3H and B6N). (**C**) Percentage of solid matter in the cecal content of five GF strains (NMRI, B6J, BALBc, C3H and B6N) (open circles (○) males, closed circles (●) females). Statistical differences were determined by one-way ANOVA with Tukey’s post-hoc test * *p* < 0.05, ** *p* < 0.01. Each dot represents one independent sample. (**D**) Average lifespan of aging cohorts of GF mice on B6J and C3H background. Data in the curve show the end-points when GF mice were sacrificed. B6J mice were sacrificed when reaching humane end-points, defined as a visibly magnified abdomen (due to the greatly enlarged cecum) that started to interfere with the locomotor activity of the animal. Mice on the C3H background were sacrificed at experimental end-points (n = 17 B6J; 8 C3H). (**E**) Percentage of water in the cecal content of complex microbiota colonized B6J and C3H mice (open circles (○) males, closed circles (●) females). Statistical differences were determined by unpaired *t*-test. Each dot represents one independent sample. (**F**) Percentage of solid matter in the cecal content of complex-microbiota colonized B6J and C3H mice (open circles (○) males, closed circles (●) females). Statistical differences were calculated by unpaired *t*-test. Each dot represents one independent sample. (**G**) Average water to solid matter in the cecal content of complex microbiota colonized B6J and C3H mice. (**H**–**K**) Untargeted metabolomic analysis of cecal content from GF B6J and C3H mice. (**H**,**I**) Principal component (PC) analysis score plots obtained in (**H**) negative and (**I**) positive ionization modes. (**J**) Heat map representation of identified metabolic features detected in positive and negative ionization mode differently expressed in cecal content of GF C3H and B6J mice (*p*  <  0.05; n = 3). (**K**) Metabolic pathway analysis plot created using MetaboAnalyst 5.0. The plot shows metabolic pathways enriched in the cecal content of GF B6J mice. The x-axis represents the pathway impact value computed from pathway topological analysis and the y-axis shows the -log10 of the *p* value obtained from pathway enrichment analysis. Abbreviations: GF, germ-free; SPF, specific pathogen free; ns, not significant; +, positive ion mode; −, negative ion mode.

**Figure 3 nutrients-15-00636-f003:**
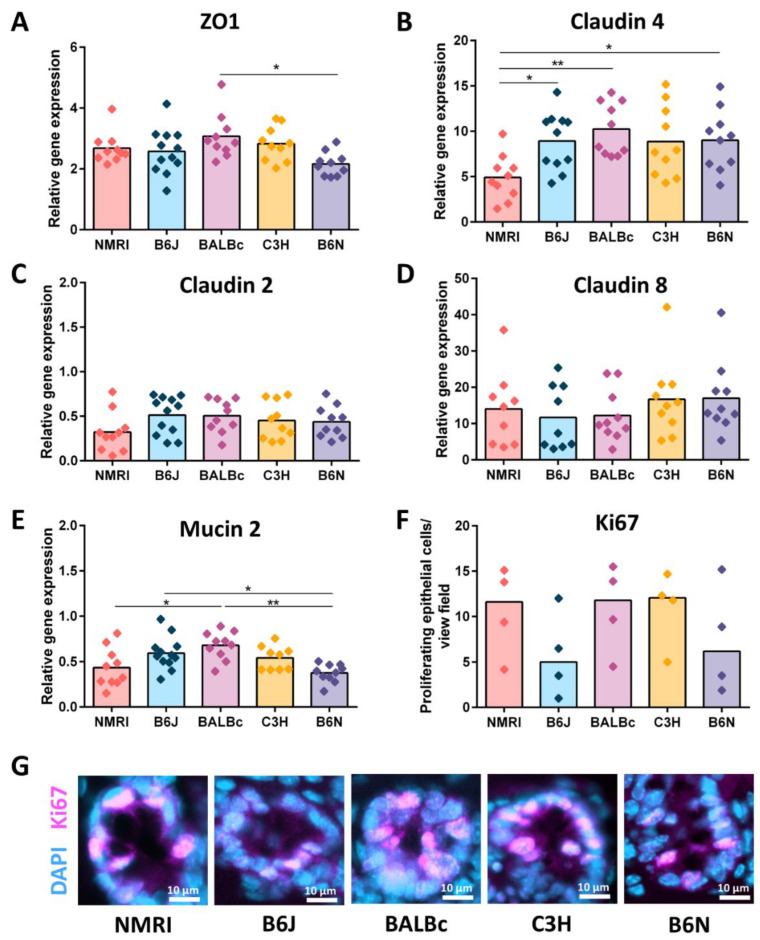
Involvement of epithelial barrier factors in cecal enlargement. (**A**–**D**) Gene expression of epithelial tight junctions: (**A**) ZO1, (**B**) Claudin 4, (**C**) Claudin 2, and (**D**) Claudin 8 measured by qPCR in total RNA isolated from the cecal tissue of five GF strains (NMRI, B6J, BALBc, C3H and B6N). Relative differences in gene expression were calculated by the comparative 2^−ΔΔCt^ method. Statistical differences were calculated by one-way ANOVA with a Tukey’s post-hoc test * *p* < 0.05, ** *p* < 0.01. Box plot indicates mean. Each symbol represents one independent sample. (**E**) Gene expression of mucin 2 measured by qPCR in total RNA isolated from cecal tissue of five GF strains (NMRI, B6J, BALBc, C3H and B6N). One-way ANOVA with Tukey’s post-hoc test * *p* < 0.05, ** *p* < 0.01. Box plot indicates mean. Each symbol represents one independent sample. (**F**) Quantification of proliferating epithelial cells (Ki67+ cells) in the cecal tissue of five GF strains (NMRI, B6J, BALBc, C3H and B6N; n = 4). Kruskal-Wallis test with Dunn´s multiple comparison test. Box plot indicates median. Each symbol represents one independent sample. (**G**) Representative immunostaining for Ki67 (violet) on cecal sections obtained from five GF strains (NMRI, B6J, BALBc, C3H and B6N; n = 4). Nuclei (light blue) were counterstained with DAPI. Scale bars 50 µm. Abbreviations: ZO-1, Zonula occludens 1; GF, germ-free.

**Figure 4 nutrients-15-00636-f004:**
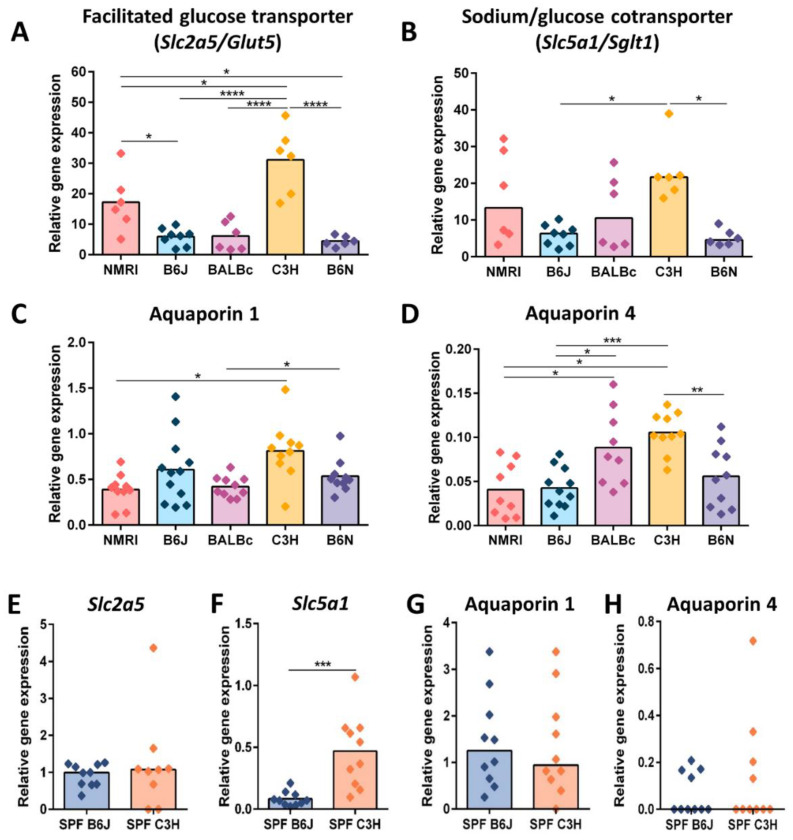
Involvement of water and electrolytes transport in cecal enlargement. (**A**,**B**) Gene expression of solute transporters: (**A**) Facilitated glucose transporter *Slc2a5/Glut5*, and (**B**) Sodium/glucose cotransporter *Slc5a1*/*Sglt1* measured by qPCR in total RNA isolated from cecal tissue of five GF strains (NMRI, B6J, BALBc, C3H and B6N). Relative differences in gene expression were calculated by the comparative 2^−ΔΔCt^ method. Statistical differences for parametric data were calculated by one-way ANOVA with Tukey’s post-hoc test * *p* < 0.05, **** *p* < 0.0001. Box plot indicates mean (Slc2a5/Glut5). A Kruskal-Wallis test with Dunn´s multiple comparison test was used for the statistical analysis of nonparametric data * *p* < 0.05. Box plot indicates median (*Slc5a1*/*Sglt1*). Each symbol represents one independent sample. (**C**,**D**) Gene expression of aquaporins: (**C**) Aquaporin 1, and (**D**) Aquaporin 4 measured by qPCR in total RNA isolated from cecal tissue of five GF strains (NMRI, B6J, BALBc, C3H and B6N). Relative differences in gene expression were calculated by the comparative 2^−ΔΔCt^ method. The statistical analysis of parametric data was performed by a one-way ANOVA with a Tukey’s post-hoc test * *p* < 0.05, ** *p* < 0.01, *** *p* < 0.001. Box plot indicates mean. Each symbol represents one independent sample. (**E**,**F**) Gene expression of glucose transporters: (**E**) Facilitated glucose transporter *Slc2a5*/*Glut5*, and (**F**) Sodium/glucose cotransporter *Slc5a1*/*Sglt1* measured by qPCR in total RNA isolated from cecal tissue of complex microbiota colonized B6J and C3H mice. Statistical differences for parametric data were calculated with an unpaired *t*-test and for nonparametric data with a Mann-Whitney test, *** *p* < 0.001. The box plot indicates mean or median. Each symbol represents one independent sample (**G**,**H**) Gene expression of aquaporins: (**G**) Aquaporin 1, and (**H**) Aquaporin 4 measured by qPCR in total RNA isolated from cecal tissue of complex microbiota colonized B6J and C3H mice. Relative differences in gene expression were calculated by the comparative 2^−ΔΔCt^ method. Statistical differences for parametric data were calculated with an unpaired *t*-test * *p* < 0.05, and for nonparametric data with a Mann-Whitney test, * *p* < 0.05. Box plot indicates mean or median. Each symbol represents one independent sample. Abbreviations: GF, germ-free; SPF, specific pathogen free.

**Figure 5 nutrients-15-00636-f005:**
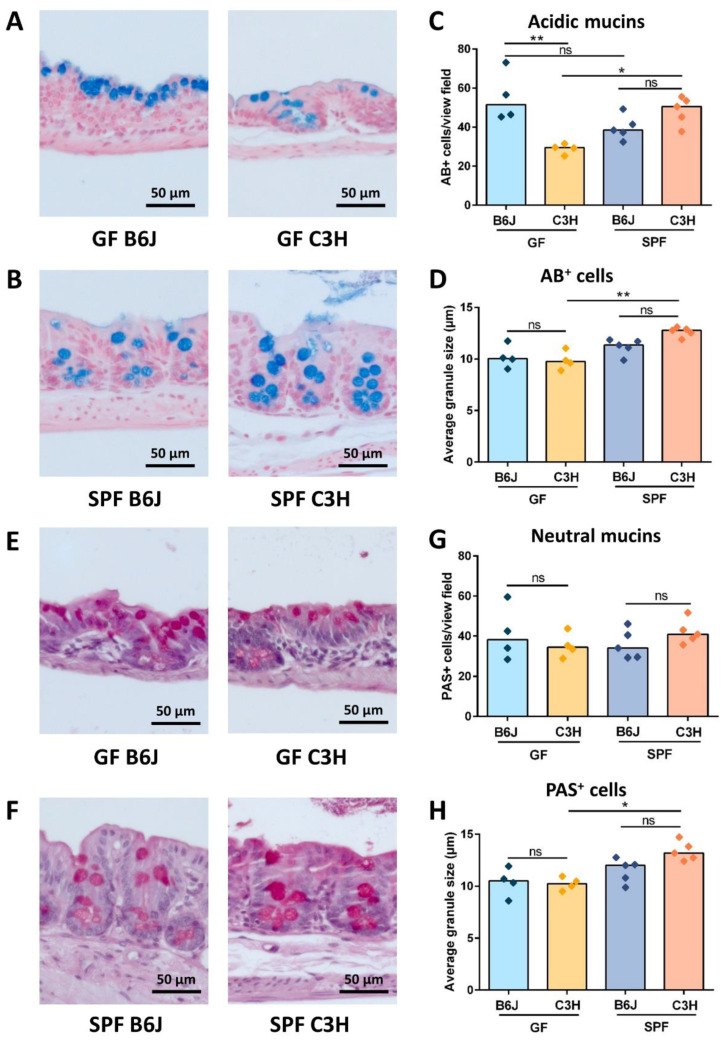
Contribution of mucins to the cecal enlargement. (**A**–**D**) Alcian blue staining: (**A,B**) Representative images of Alcian blue staining of (**A**) GF B6J and C3H and (**B**) complex microbiota colonized B6J and C3H cecal tissue. Acidic mucin-secreting goblet cells are stained blue. Nuclei (light red) and background (pale pink) are counterstained with nuclear fast red. (**C**) Quantification of Alcian blue positive cells in the cecal tissue of GF and complex microbiota colonized B6J and C3H mice (n = 4; 5). A statistical analysis was performed by a Kruskal-Wallis test with Dunn´s multiple comparison test * *p* < 0.05, and ** *p* < 0.01. Box plot indicates median. Each symbol represents one independent sample. (**D**) Average granule size of Alcian blue stained cells in the cecal tissue of GF and complex microbiota colonized B6J and C3H mice (n = 4; 5). A statistical analysis was performed by a Kruskal-Wallis test with Dunn´s multiple comparison test ** *p* < 0.001. Box plot indicates median. Each symbol represents one independent sample. (**E**–**H**) Periodic acid-Schiff (PAS) staining: (**E**,**F**) Representative images of PAS staining of (**E**) GF B6J and C3H and (**F**) complex microbiota colonized B6J and C3H cecal tissue (n = 4; 5). Neutral mucin-secreting goblet cells are stained pink to red. Nuclei are stained blue to violet. (**G**) Quantification of PAS positive cells in the cecal tissue of GF and complex microbiota colonized B6J and C3H mice (n = 4; 5). Statistical analysis was performed by a Kruskal-Wallis test with a Dunn´s multiple comparison test. Box plot indicates median. Each symbol represents one independent sample. (**H**) Average granule size of PAS stained cells in the cecal tissue of GF and complex microbiota colonized B6J and C3H mice. A statistical analysis was performed by a Kruskal-Wallis test with a Dunn´s multiple comparison test * *p* < 0.05. Box plot indicates median. Each symbol represents one independent sample. Abbreviations: GF, germ-free; SPF, specific pathogen free; ns, not significant; AB+, Alcian blue positive; PAS+, periodic acid-Schiff positive.

**Figure 6 nutrients-15-00636-f006:**
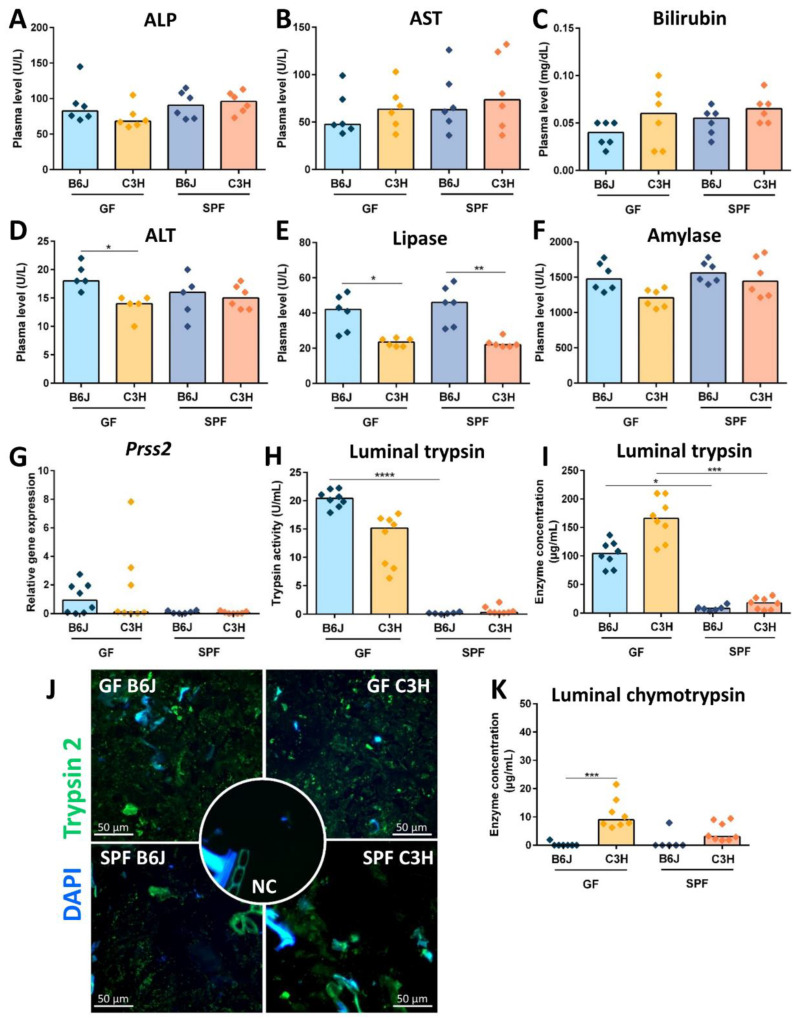
**Contribution of digestion enzymes to the cecal enlargement.** (**A**–**D**) Liver enzymes: (**A**) alkaline phosphatase (ALP), (**B)** aspartate aminotransferase (AST), (**C**) bilirubin, and (**D**) alanine aminotransferase (ALT) determined in plasma of GF and complex microbiota colonized B6J and C3H mice (n = 6). Kruskal-Wallis test with Dunn´s multiple comparison test * *p* < 0.05. Box plot indicates median. Each symbol represents one independent sample. (**E**–**G**) Pancreatic digestive enzymes: (**E**) pancreatic lipase, (**F**) pancreatic amylase determined by routine clinical chemistry methods in plasma of GF and complex microbiota colonized B6J and C3H mice (n = 6). Kruskal-Wallis test with Dunn´s multiple comparison test ** *p* < 0.001, * *p* < 0.05. Box plot indicates median. Each symbol represents one independent sample. (**G**) Gene expression of *Prss2* measured by qPCR in total RNA isolated from the pancreas of GF and complex-microbiota colonized B6J and C3H mice. Relative differences in gene expression were calculated by the comparative 2^−ΔΔCt^ method. Kruskal-Wallis test with Dunn´s multiple comparison test. Box plot indicates median. Each symbol represents one independent sample. (**H**–**K**) Luminal levels of digestive enzymes: (**H**) Luminal trypsin activity of GF and complex microbiota colonized B6J and C3H mice measured by ELISA. Kruskal-Wallis test with Dunn´s multiple comparison test **** *p* > 0.0001. Box plot indicates median. Each symbol represents one independent sample. (**I**) Concentration of luminal trypsin measured in cecal content of GF and complex microbiota colonized B6J and C3H mice. Kruskal-Wallis test with Dunn´s multiple comparison test *** *p* > 0.001, * *p* > =0.05. Box plot indicates median. Each symbol represents one independent sample. (**J**) Representative immunostaining for trypsin 2 (green) of colon lumen obtained from GF and complex microbiota colonized B6J and C3H mice (n = 5) including staining negative control (NC). Colon slides were counterstained with DAPI (blue). (**K**) Concentration of luminal chymotrypsin measured in cecal content of GF and complex microbiota colonized B6J and C3H mice. Kruskal-Wallis test with Dunn´s multiple comparison test *** *p* > 0.001. Box plot indicates median. Each symbol represents one independent sample.

## Data Availability

The authors confirm that the data supporting the findings of this study are available within the article, its Appendix A or are openly available in MassIVE under ID MSV000090765.

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
