# Peer review of "The Genetic Background Is Shaping Cecal Enlargement in the Absence of Intestinal Microbiota"

_nutrients, 2023, doi:10.3390/nu15030636_

Round 1

Reviewer 1 Report

The manuscript by M. Basic and colleagues investigates the role of the genetic background in the enlargement of the cecum observed in mice reared under germ-free (GF) conditions. The authors use 5 different wild type mouse strains common in rodent research to ask whether they exhibit cecal enlargement in GF conditions and to what extent. Then, they are trying to understand the differences observed between strains by addressing expression of genes involved in epithelial junction formation and water transport and absorption, as well as function of digestion enzymes. They also compare the strains exhibiting the most extreme phenotypes of cecal size using an untargeted metabolomic approach of their ceca to identify specific metabolites that may be contributing to/explaining the GF phenotypes. Their results clearly show that the genetic background of different mouse strains contributes to the cecal enlargement phenotype in GF conditions, which is caused mainly by water retention in the gut lumen. The authors associate this phenotype with decreased expression of water transporters and increased production of acidic mucins in the gut. Their mass spectrometry metabolite analysis underscores distinct metabolic cecal profiles of the strains exhibiting extreme cecal size phenotypes. The study provides a thorough description of the differences of various wild type strains with regards to GF cecum size, and some interesting associations with genes and metabolites implicated in intestinal water homeostasis. It will be useful for future rodent GF research studies. 

Specific comments for the improvement of the manuscript:

1.     Page 7: what is the body weight of SPF B6J and C3H mice? A graph in Fig. 1 would help. 

2.     Fig. 1: in GF conditions, the body weight and cecal weight vary in females vs. males and this sex difference is very pronounced in B6J mice. This is not the case in SPF conditions. Do the authors have an explanation for this? 

3.     Line 352: the title could be rephrased to “Host genetic background shapes distinct metabolic profiles in the GF mouse cecum” to be more accurate.

4.     Figures and figure panels should be described in the text in the order they appear. For example, on page 9: reference of Fig. 2B should precede Fig. 2C (lines 359, 361); on page 11: Figure S4 (line 416) is described before Fig. S3 (line 421); on page 12: Fig. S1 is described after Fig. S2-S4. Please check throughout the manuscript.

5.     Fig. 2J: Genistein and Tauro-a-muricholic acid appear twice in the heat map. Why?

6.     Page 11, line 415: in Fig. 2J the authors show 22 rows/metabolites in the heat map (with 2 metabolites shown twice, see point 6 above) and in line 415 they mention the same number of 22 metabolites as differentially abundant between strains. But then in Fig. S2 and Fig. S4C the plots shown are 16 (with 2 plots for Genistein) and 5, respectively. The data shown in Fig. 2J and relevant supplements should agree and the appearance of metabolites twice in the heat map should be justified.

7.     Lines 425 and 432: replace “expression” with “abundance”. Since the metabolic profile measures metabolite quantity and not expression (which usually implies gene expression) the term abundance describes the differences more accurately.

8.     Line 437, 686, 687: “feed-derived” should be replaced by “food-derived”.

9.     Section 3.3: are tight junction proteins and Mucin 2 shown to be regulated at the level of transcription? Wouldn’t staining for these proteins be more informative to assess the epithelial integrity of the cecum?

10.  Fig. 3F: cell proliferation is measured by Ki67 staining in the ceca of different strains. What is the n number of ceca/images assessed? In the methodology (section 2.6), it is stated that 10 fields are used, but in Fig. 3F only 4 points are shown per strain. Are the observed differences statistically significant?

11.  Fig. 5: According to the scale bars, the images in B,E,H,K panels are of the same magnification. It is interesting to note that the cecal epithelium seems thicker in the SPF (H,K) compared to the GF animals (B,E) irrespective of the genotype, and also the mucin-positive goblet cells are larger in SPF ceca. Are there more cells in the SPF cecum? Is there more deposition of ECM components? 

12.  Mucin 2 expression is not altered between GF B6J and C3H ceca (Fig. 3E), but a difference in the number of (acidic) mucin-producing goblet cells is observed between the strains (Fig. 5). One would think that there would be a correlation between the two phenotypes.

Reviewer 2 Report

The work entitled ‘The genetic background is shaping cecal enlargement in the absence of intestinal microbiota’ by Silvia Bolsega et al. is very carefully prepared and interesting. It raises the currently important topic of intestinal microbiota in a slightly different aspect. I have two remarks.

I would strongly suggest removing the animal images, leaving only the cecum images, so as not to unnecessarily shock by the potential animal suffering of opponents of animal research. You can then put a scale on the photos of the cecum, to show the cecal size to the reader.

Why were  GF C57BL/6JZtm (B6J), C57BL/6NRjZtm (B6N), 67 NMRI/MaxZtm (NMRI), Balb/cJZtm (BALBc), and C3H/HeOuJ (C3H) models chosen? I am missing an explanation of this fact in the manuscript. Please complete this missing information.
